# Shanghai Fever: Not Only an Asian Disease

**DOI:** 10.3390/pathogens11111306

**Published:** 2022-11-07

**Authors:** Claudia Colomba, Michela Scalisi, Valeria Ciacio, Chiara Albano, Sara Bagarello, Sebastiano Billone, Marco Guida, Salvatore Giordano, Laura A. Canduscio, Mario Milazzo, Salvatore Amoroso, Antonio Cascio

**Affiliations:** 1Division of Pediatric Infectious Diseases, “G. Di Cristina” Hospital, ARNAS Civico Di Cristina Benfratelli, 90100 Palermo, Italy; 2Department of Health Promotion, Maternal and Infant Care, Internal Medicine and Medical Specialties, University of Palermo, 90100 Palermo, Italy; 3Pediatric Surgery Unit, “G. Di Cristina” Hospital, ARNAS Civico Di Cristina Benfratelli, 90100 Palermo, Italy; 4Infectious and Tropical Diseases Unit, AOU Policlinico “P. Giaccone”, 90100 Palermo, Italy

**Keywords:** Shanghai fever, *Pseudomonas aeruginosa*, ecthyma gangrenosum, necrotizing enteritis, sepsis

## Abstract

Objectives: To describe a case of Shanghai fever disease and to analyze other published reports in non-Asiatic countries, defining clinical characteristics and highlighting that this is not only an Asian disease. Study design: A computerized search without language restriction was conducted using PubMed and Scopus; all references listed were hand-searched to identify any other relevant literature. An article was considered eligible for inclusion in the systematic review if it reported cases with Shanghai fever described in non-Asiatic countries. Our case was also included in the analysis. Results: Ten articles reporting 10 cases of Shanghai fever disease were considered. Fever, diarrhea and ecthyma gangrenosum were the most frequent symptoms observed. Blood was the most common site of isolation for *Pseudomonas aeruginosa*. Three patients underwent surgery due to necrotizing enteritis and intestinal perforation. Meningitis was documented in one case. None of the patients received antipseudomonal antibiotics within 24 h of admission. The outcome was good in nine cases; only one patient died due to multiple organ failure from *Pseudomonas* sepsis. No common primary immune deficiency was identified in these patients. Extremely young age (<1 year) was the only host factor predisposing to Shanghai fever. Conclusions: It is important to shed light on this disease in non-Asiatic countries and take into account that it can also affect healthy children. Pediatricians, therefore, should consider Shanghai fever among diagnoses in children with community-onset diarrhea, fever and skin lesions suggestive of ecthyma gangrenosum to start an appropriate treatment sooner and to reduce the mortality in these children.

## 1. Introduction

Shanghai fever is a rare sepsis-associated enteric disease caused by *P. aeruginosa,* described mainly in previously immunologically healthy children. To our knowledge, only one case has been described in the literature in an adult patient with neutropenia. The disease was reported as early as 1918 and has been more recently defined by Chuang et al. with three criteria: (1) community-onset diarrhea with fever, (2) sepsis and (3) growth of *P. aeruginosa* from blood or another sterile body site. Necrotizing enteritis is the major complication of Shanghai fever and mortality reported worldwide is high [1,2].

To date, most of the reported cases regarding Shanghai fever are restricted to East Asian countries and only very few cases are described to be from North America and Europe.

We report a case of Shanghai fever in a previously healthy Italian 7-month-old boy and analyze the medical literature to report the epidemiologic and clinical characteristics of cases of Shanghai fever from non-Asiatic countries.

## 2. Methods

This systematic review was performed in accordance with the PRISMA protocol (Reporting Items for Systematic Reviews and Meta-Analyses) and a systematic review registration is currently ongoing at PROSPERO.

A computerized literature search was performed on PubMed and Scopus search engines by submitting the query (Pseudomonas AND (enteritis OR diarrhea OR colitis OR enterocolitis) AND (sepsis OR septicemia OR bacteriemia) AND (children OR child OR baby OR infant)) OR Shanghai fever (title/Abstract). No filters or language restrictions were applied to the results. Furthermore, all references listed were hand-searched for other relevant articles, and a citation tracker was used to identify any other relevant literature. An article was considered eligible for inclusion if reporting cases with full clinical data consistent with Shanghai fever. The following epidemiologic and clinical variables were evaluated for each case: gender, age, clinical manifestations, intestinal involvement, antibiotic therapy, surgical procedures and clinical outcome. Outcome was considered good in patients responding to treatment and not presenting with sequelae.

## 3. Results

From the literature search carried out we retrieved 50 papers, of which only 2 were eligible [3,4]. In particular, we excluded from the analysis eleven papers because they were describing cases related to Asian countries and eight because they involved immunosuppressed patients. Other papers were not relevant. In a manual search of the bibliography, we found seven relevant articles (Figure 1).

Nine articles, each reporting single cases of Shanghai fever, were eventually included, dating from 1980 to 2022 [3,4,5,6,7,8,9,10,11]. All cases in our review meet the criteria of Shanghai fever [2]. Our review reports cases from non-Asiatic countries and, in particular, four cases from Europe, four cases from North America, one case from South America and one case from South Africa. The median age of the patients was 7 months and 80% were aged <1 year. The male-to-female ratio was 6:4. The most common clinical manifestations were fever (100%), diarrhea (100%) and ecthyma gangrenosum (90%). All patients with ecthyma gangrenosum had multiple sites affected. One patient had meningitis [9]. The median time from onset of the first symptom to sepsis was four days (range: three to six days). Three had bowel perforation requiring immediate surgical intervention [5,10]. Intraoperative findings showed widespread patchy necrosis with fibrin coating of the small intestine or colon. No patients had short bowel syndrome after surgery.

Leukopenia, thrombocytopenia, high C-reactive protein (CRP) levels, coagulopathy and hypoalbuminemia were the characteristic laboratory findings. Leukopenia and thrombocytopenia were found in six cases (60%), and leukocytosis in 2 cases (20%). Mild to moderate anemia was found in eight cases (80%) described in our review.

Blood was the most common site of isolation for *P. aeruginosa* (70%). *P. aeruginosa* was also isolated from skin lesions (60%), stool (50%), peritoneal fluid (30%), tracheal or bronchial secretions (30%), urine samples (20%), oropharyngeal exudate (20%), cerebrospinal fluid (10%) and pleural effusion (10%).

None of the patients received antipseudomonal antibiotics within 24 h of admission. In all cases, antipseudomonal therapy was started after a positive cultures result. Only one patient died due to multiple organ failure from *Pseudomonas sepsis*. In this case, the baby was hospitalized on the fifth day of her illness in poor condition (distributive shock and pancolonic necrosis) [5].

Taken together, no definitive common primary immune deficiency was identified in these patients based upon both laboratory and clinical evaluations. Extremely young age (<1 year) was the only identified host factor predisposing to Shanghai fever.

## 4. Case Report

A previously healthy 7-month-old male child was admitted with three days of high graded fever, hyporexia and diarrhea associated with perianal erythema. On admission, his body temperature was 37.7 °C, respiratory rate was normal and heart rate was 140 beats/min. An examination revealed pharyngeal hyperemia without exudate, two small erythematosus lesions on the legs initially interpretated as insect bites, de-epithelialization and erythema of the perianal skin. The abdomen was soft and painless and no signs of meningeal irritations could be detected during the inspection. The first laboratory workup revealed leukopenia (white blood cells 2.8 × 10^3^/μL, neutrophils 1.38 × 10^3^/μL and lymphocyte count 1.2 × 10^3^/μL), mild anemia (Hb 10.7 g/dL), platelets 63 × 10^3^/μL, high inflammatory markers (CRP 27.3 mg/dL, procalcitonin 70 μg/L), hypoalbuminemia, hyponatremia and hypokalemia. Liver enzymes, renal function tests, prothrombin time and activated thromboplastin time were always normal with a subsequent increase of D-dimer. The suspicion of sepsis led to the decision of performing a blood culture and to start an intravenous antibiotic therapy with Ceftriaxone. On the second day of hospitalization, we observed a worsening of clinical conditions. The patient presented signs of meningeal irritation such as neck stiffness and photophobia, in addition to continuous and mournful crying. He had developed whitish tonsillar exudate, foul-smelling breath and multiple hyperemic and infiltrated skin lesions over the legs and buttocks, which were rapidly becoming ulcerative-necrotizing skin lesions suggestive of ecthyma gangrenosum (Figure 2).

Perianal lesions became worse with the loss of substance covered in greenish secretions. A lumbar puncture and microbiological analysis of cerebrospinal fluid were performed. *Escherichia coli* and *P. aeruginosa* were isolated from blood cultures. *Pseudomonas* colonies were also isolated from skin lesions, oropharyngeal exudate, stool sample and bronchial aspirate culture. The results from the urine culture and the microbiological analysis of cerebrospinal fluid were within the normal range. The treatment was switched to antipseudomonal antibiotic by starting a therapy with Meropenem. A substantial decrease of CRP and procalcitonin levels, as well as an increase in the number of leukocytes, were registered in the patient since then.

On the fifth day of admission, the patient presented fever associated with bilious vomiting, severe abdominal distension and peritonism. An abdominal X-ray showed signs of hollow organ perforation.

Laparotomy revealed two bowel perforations and widespread patchy necrosis with fibrin coating across the entire small intestine (Figure 3).

An intestine tract of about 7 cm was resected and an ileostomy was performed. *P. aeruginosa* was also isolated from peritoneal fluid.

During an extensive immunological study performed to the aim of excluding any primary immune deficiency disease, the immunological evaluation of the patient showed normal lymphocyte subset counts and CD11/CD18 expression, normal complement and immunoglobulin (Ig) levels (IgG 591 mg/dL, IgA 45.2 mg/dL, IgM 61.9 mg/dL), negative HIV test and normal oxidative burst in the dihydrorhodamine test (DHR123).

Over about a month after the admission, the patient showed an improvement of clinical conditions and skin lesions, which led to his discharge from the hospital. Negative blood culture, skin lesions culture and stool culture were obtained before the patient left the hospital.

Nineteen days after his discharge, the baby was hospitalized again because of bilious vomiting caused by two intestinal obstructions secondary to adhesion formations. He was operated upon again and dismissed from hospital five days after the intervention.

## 5. Discussion

*P. aeruginosa* is an opportunistic, aerobic Gram-negative bacterial pathogen responsible for a variety of genitourinary, pulmonary, skin and soft tissue infections in hospitalized pediatric patients, often in association with significant morbidity [12,13,14,15]. Rarely, *P. aeruginosa* can lead to severe and life-threatening infections among previously healthy children [16,17,18,19]. Skin lesions such as subcutaneous nodules and ecthyma gangrenosum may be the first manifestation of *Pseudomonas* sepsis that have rarely been reported even in healthy children [5,20,21,22,23]. *P. aeruginosa* can also affect the entire gastrointestinal tract. Hundreds of cases of Shanghai fever have been reported in literature, mostly from East Asia, and with very few exceptions from North America and Europe. All the cases reported in this review meet the criteria of Shanghai fever, although they were registered outside East Asian countries. Chuang et al. described the biggest case series of Shanghai fever in 27 patients in Taiwan [2]. The clinical manifestations of *P. aeruginosa* sepsis among patients from Eastern countries appear to be different from the ones reported in Western countries. Fever and diarrhea are the most common presentations in East Asia, whereas fever and skin lesions are more common in cases reported from North America and Europe [2]. Ecthyma gangrenosum was reported in 90% of the cases studied in our review, but only in 63% of the patients described by Chuang. The identification of ecthyma gangrenosum lesions could be useful to the earliest possible introduction of the antipseudomonal therapy, since empiric antibiotic treatment commonly used in sepsis in the immunocompetent child are not effective for Pseudomonas infections [18].

In cases of Shanghai fever in East Asia, most patients have necrotizing enteritis, while bowel perforation is reported in 30% of cases. In our review of cases from complementary regions, three of the patients (30%) have undergone surgery for intestinal perforation.

Necrotizing-ulcerative lesions can also involve the oral cavity and these lesions may lead to perforation of the hard and soft palate, as in one of the cases reported in our review in which the debridement of palatal necrotic tissues and closure of soft palate defect was necessary. In the same case, ecthyma gangrenosum evolved into leg gangrene [24].

Seizures are frequently associated with Shanghai fever and meningitis is the most serious complication of the central nervous system [2]. The patient in our case study had signs of meningeal irritation, such as neck stiffness, photophobia and drowsiness, but the cerebrospinal fluid analysis resulted negative. Only in one of the cases reported in our review *P. aeruginosa* was isolated in cerebrospinal fluid [9], while CNS depression and neck stiffness were described in two other cases [4,6].

We reported the most important symptoms of Shanghai fever in non-Asiatic countries in Table 1.

Furthermore, our study seems to confirm existing observations on Shanghai fever, stating that infants account for >80% of reported cases [2].

Neutropenia is one of the main risk factors for the development of ecthyma gangrenosum and severe *Pseudomonas* infections. It has been suggested that infection with *Pseudomonas* in healthy children can cause a transient neutropenia by producing a toxin that decreases the number of neutrophils in circulation, such as in patients with Shanghai fever and no immune deficiency disease [18,23]. Five patients (50%) in our review had transient neutropenia on admission and the young age appears to be the only risk factor identified in the host. None of the patients in our review presented a primary or acquired immune deficiency disease.

As shown by Chuang et al., Shanghai fever is not associated with hospital-acquired colonization but is caused by more virulent community strains of *P. aeruginosa*, demonstrating a higher cytotoxic and invasive profiles [2]. *P. aeruginosa* can present various antibiotic resistant mechanism, including the formation of biofilms, the production of virulence toxins and enzymes, low membrane permeability and the expression of antibiotic efflux pumps [24,25]. Unfortunately, we were unable to perform molecular analysis or virulence assays in our isolate, but the *P. aeruginosa* we isolated was a biofilm producer microorganism. Environmental inspections were performed by the family and *P. aeruginosa* was isolated in their domestic water supply. In one case considered by this review, *P. aeruginosa* was isolated in samples of bottled water used to prepare baby formulas [7].

Our case study is compatible with Shanghai fever syndrome despite the epidemiological profile of the latter. Since the patient had no history of traveling outside Italy during his life, it is possible to affirm that more virulent strains of *P. aeruginosa* are not restricted to Asian countries and may cause Shanghai fever in Italy.

In conclusion, we believe it is important to shed light on this syndrome in non-Asiatic countries and to take into account that it can also affect healthy children. Its clinical features are younger age, especially infancy, with a bloody, mucoid or greenish stool pattern. Diagnosis should be rapid in order to begin a timely treatment with antipseudomonal antibiotics.

## Figures and Tables

**Figure 1 pathogens-11-01306-f001:**
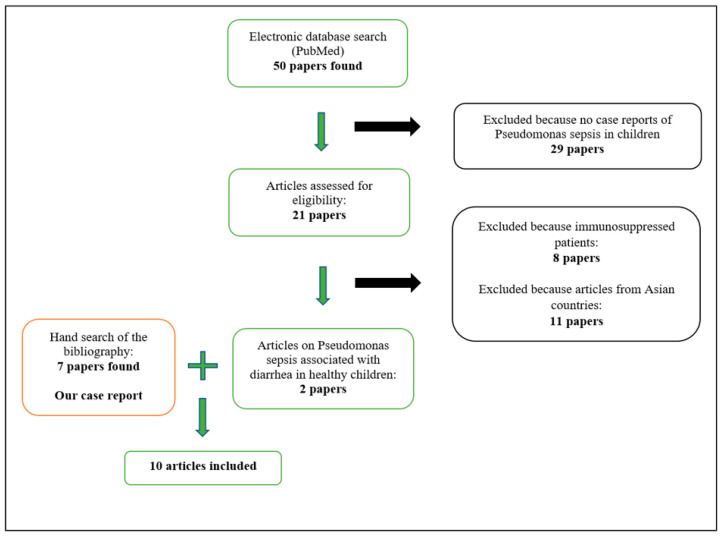
Our search strategy for the systematic review.

**Figure 2 pathogens-11-01306-f002:**
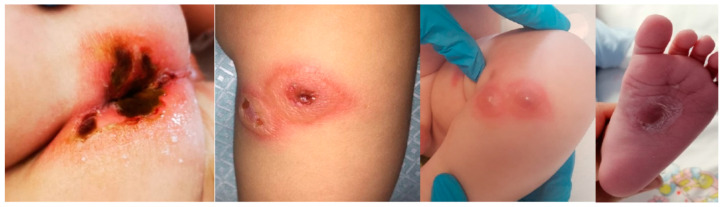
Hyperemic and infiltrated skin lesions over the legs and buttocks turning into ulcerative-necrotizing skin lesions suggestive of ecthyma gangrenosum.

**Figure 3 pathogens-11-01306-f003:**
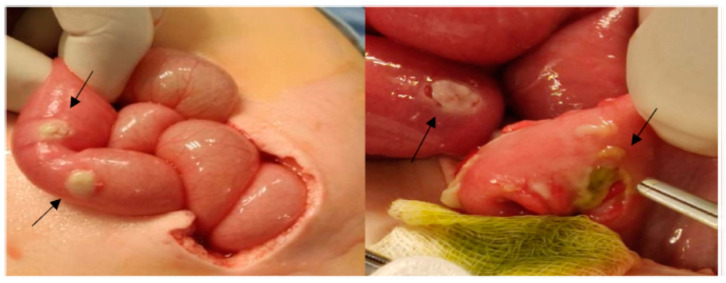
Widespread patchy necrosis with fibrin coating across the entire small intestine.

**Table 1 pathogens-11-01306-t001:** The most important symptoms of Shanghai fever reported in non-Asiatic countries.

Most Important Symptoms of Shanghai Fever	Number (%)
Fever	10 (100)
Diarrhea	10 (100)
Ecthyma gangrenosum	9 (90)
Shock	6 (60)
Dyspnoea/cyanosis	5 (50)
Vomiting	4 (40)
Hepatomegaly/hepatosplenomegaly	3 (30)
Necrotising enteritis/Bowel perforation	3 (30)
Meningitis/meningism	3 (30)

## Data Availability

All data used and/or analyzed during this study are included in this article.

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
