# Peer review of "Shanghai Fever: Not Only an Asian Disease"

_pathogens, 2022, doi:10.3390/pathogens11111306_

Round 1
Reviewer 1 Report
This review article entitled "Shanghai Fever: Not Only an Asian Disease" reported by Colomba et al., which presented cases from Shanghai fever from Italy. As authors have written, Shanghai fever is supposed to be very rare in non-Asian countries. I believe this case report is clinically important for clinicians, so that it is worth publishing in Pathogens.However, there are some points to be considered in this manuscript. Please respond following points.
Major comments
1. Authors defined Shanghai fever as ‘‘an enteric disease in previously immunologically healthy children.’’ However, there is a report of Shanghai fever occurred in adult patients. Authors should correct the definition.
2. Authors mentioned that the clinical manifestations of P. aeruginosa sepsis among patients from Eastern countries appear to be different from the ones reported in Western countries. Are there any possible cause of difference of clinical manifestations between the two regions ?
Minor points
1. Authors should put arrows near the necrotic area in Figure 3 to increase the impact.
Reviewer 2 Report
This article should present as 'Case Report' in appropriate journal.
Reviewer 3 Report
The article is the connection of the review and the case report. Worldwide are many diseases and / or infections, which are observed very rare. So disease is Shanghai fever, which according to the reviewed article, was described in 9 articles from 1980 to 2022. I suggest only one correction. I would like to ask for adding of a scheme or figure in the Results, with a presentation of the most important / often symptoms of Shanghai fever.
Reviewer 4 Report
In general, it is a good review that reported 10 cases of shanghai fever from non-asiatic countries, indicating that shanghai fever is not only an asian disease, which I agree with the authors on. The authors evaluated the epidemiologic and clinical variables for 9 cases and then specifically focused on an Italian case where a healthy 7-month-old boy got infected by this bacterial pathogen and detailed clinical manifestations and laboratory findings were assessed. This manuscript can be accepted if the following questions could be addressed properly:
1. Is the pathogen that caused this disease both in Asia and non-asian countries the same strain? If not, how many stains have been found? Which one(s) are more virulent over the others? Can you provide any genetic information regarding different strains, for example, where the mutations occur? And how is it associated with age, symptoms, and recovery time? The authors briefly mentioned it in the discussion section, but more clarification is needed.
2. In terms of treatment, do they use the same antibiotics for both asian and non-asian countries? Since antibiotic resistance always happens, has any potential new drug been proposed in any literature? what might be its mechanism overcoming drug resistance?
3. Although shanghai fever is not limited to East Asia, why are there much more cases in Asia than in other countries? Any explanations?
4. Minor language issues including spelling need to be fixed. For example, line 147, "let" should be "left". Line 173, "Most patients with shanghai fever have.....from East Asia" also need to be fixed grammatically.
Round 2
Reviewer 2 Report
As this paper clinically important, it should better suited for the 'Clinical Case' based Journal. However, it can be accepted!